# Risk Factors for Difficult Peripheral Intravenous Cannulation. The PIVV2 Multicentre Case-Control Study

**DOI:** 10.3390/jcm9030799

**Published:** 2020-03-15

**Authors:** Miguel Angel Rodriguez-Calero, Joan Ernest de Pedro-Gomez, Luis Javier Molero-Ballester, Ismael Fernandez-Fernandez, Catalina Matamalas-Massanet, Luis Moreno-Mejias, Ian Blanco-Mavillard, Ana Belén Moya-Suarez, Celia Personat-Labrador, José Miguel Morales-Asencio

**Affiliations:** 1Nurse Director Office, Health System of the Balearic Islands (Ib-Salut), Carrer de la Reina Esclaramunda, 9. Piso 3, 07003 Palma Mallorca, Spain; 2Department of Nursing and Physiotherapy, University of the Balearic Islands (UIB), Ctra Palma-Valldemossa km 7.5, 07121 Palma Mallorca, Spain; depedro@uib.es (J.E.d.P.-G.); ianblanco@hmanacor.org (I.B.-M.); celia.personat@uib.es (C.P.-L.); 3Balearic Islands Health Research Institute (IdISBa), Ctra Valldemossa, 79. Hospital Universitari Son Espases, edifici S, 07120 Palma de Mallorca, Spain; 4Hospital Manacor, Ctra Manacor-Alcudia s/n, 07500 Manacor (Mallorca), Spain; ljmolero@hmanacor.org (L.J.M.-B.); isfernandez@hmanacor.org (I.F.-F.); cmatamalas@hmanacor.org (C.M.-M.); lmoreno@hmanacor.org (L.M.-M.); 5Costa del Sol Hospital, Endoscopy & digestive medicine unit. Autovia A-7, Km. 187, 29603 Marbella, Spain; celia.personat@uib.es; 6Instituto de Investigación Sanitaria de Málaga (IBIMA), Universidad de Málaga, C/ Arquitecto Francisco Peñalosa, 3, 29071 Málaga, Spain; jmmasen@uma.es

**Keywords:** peripheral venous catheterization, risk factors, cardiovascular system, hospitalization

## Abstract

Background. Difficult peripheral intravenous cannulation (DPIVC) is associated with serious complications related to vascular access. These complications might be avoided if the risk factors were identified previously, enabling the detection of potentially difficult situations at an early stage. The aim of this study is to consider these risk factors, to determine the influence of the hospital setting, to examine the association between DPIVC and the different techniques of catheter insertion and to analyse the importance of the clinician’s experience in this context. Methods. Case-control study following a previously published protocol, conducted in 48 units of eight public hospitals in Spain. Adult patients requiring a peripheral intravenous cannula were prospectively included in the study population during their hospital stay. Over a period of 11 months, for consecutive eligible patients, nurses in each participating unit recorded data on their assessment of the vascular access performed and the technique used. Variables related to these medical personnel were also recorded. One of the researchers reviewed the patients’ clinical history to compile the relevant health variables and to characterise the healthcare process. The statistical analysis included association tests among the main study variables. The risk factors were analysed using bivariate logistic regression. The variables found to be statistically significant were included in a multivariate logistic regression model incorporating each of the healthcare environments identified. Results. The study population was composed of 2662 patients, of whom 221 (8.3%) presented with DPIVC. A previous history of difficulty, the presence of non-palpable veins, acute upper limb alterations and punctures in the ante-cubital fossa were found to be independent risk factors for DPIVC. Differences were found in the frequency of occurrence of DPIVC and in some risk factors, according to the healthcare context. The variables related to the characteristics of the hospital personnel did not influence the study event. Conclusion: The present study identifies four independent risk factors for DPIVC that can be incorporated into algorithms aimed at preventing its occurrence and facilitating the referral of patients to vascular access specialist teams.

## 1. Introduction

Peripheral intravenous catheters (PIVC) are the most commonly used invasive devices in hospital care. Around 60% of patients treated in hospital, and up to 90% of those treated in hospital emergency rooms, receive a PIVC for the administration of fluids and drugs or to obtain blood samples [1].

On numerous occasions, several punctures must be performed in order to achieve successful catheterization. These successive attempts can produce pain and delay the start of diagnostic treatments or tests [2]. Furthermore, repeated punctures can degrade vascular walls, complicating subsequent approaches, and reducing the quality of healthcare [3]. This situation has been described as ‘difficult intravenous access’ (DIVA) [4,5], or as ‘difficult peripheral intravenous cannulation’ (DPIVC) to refer specifically to the difficulty in obtaining peripheral vascular access [6].

To date, there is no consensus on the most appropriate definition of DPIVC, but it generally refers to situations of two or more failed cannulations and/or the need for advanced or rescue techniques [7].

DPIVC has been reported to affect 10%–24% of hospitalized adults and is associated with higher rates of catheter failure, provoking more frequent catheter replacement [8]. This may produce serious complications associated with vascular access, such as phlebitis, extravasation, bruising, haemorrhages, catheter-associated infection and sepsis, thus aggravating morbidity [9,10].

Frequent catheter replacement severely degrades the vascular tree, and can lead to a situation in which no available locations remain for puncture, known as ‘vascular exhaustion’. In these circumstances, it is necessary to resort to central access catheterization, not because of clinical need, but as a rescue option [11]. However, the use of central access vascular devices is also associated with increased access-related morbidity and mortality. This outcome could be avoided if it were possible to improve the effectiveness of the first peripheral puncture and insert the most suitable catheter at the start of intravenous treatment [12,13]. In this regard, it has been shown that the need for central catheterization can be reduced by 74%–85% by implementing programmes based on advanced cannulation techniques, thus reducing access-related complications and achieving cost savings of 220–1600 US dollars for each catheter inserted [14,15].

Knowledge of the risk factors for DPIVC would enable the identification of high-risk patients at an early stage and their referral for advanced insertion techniques, thus avoiding the repetition of punctures and their negative consequences, facilitating the selection of the most appropriate vascular device according to the patient’s clinical situation, improving the effectiveness of IV treatment and reducing costs arising from complications and repetition [15].

At least 60 possible risk factors have been proposed for DPIVC among adult patients, including demographic variables (age, gender, ethnicity, etc.), anthropometric values (BMI, height, etc.), conditions of vascular access (visibility and palpability of the target vein, vessel diameter, previous history of difficulty with punctures or in the insertion of catheters), patient’s health status (diabetes mellitus, cancer, parenteral drug abuse, etc.), variables related to healthcare (recent hospital care, active treatment with chemotherapeutic drugs, etc.) and variables related to the medical personnel performing the technique (clinical experience, evaluation of the access situation) [16,17].

Prior studies in this field have analyzed only limited elements among these factors, in specific hospital areas, mainly accident and emergency (A&E), surgical areas, radiology facilities or hospitalization units. The heterogeneity of prior research efforts and the limitations presented in each case highlight the need for a single study to be undertaken exploring a wide range of potential risk factors, including different profiles within the hospital population, and determining the specific weight to be assigned to each factor [17].

Accordingly, the main aim of this study is to identify the independent risk factors of DPIVC associated with the patient. As a secondary objective, the study also characterizes the patients most at risk of experiencing DPIVC according to specific treatment profile (A&E, surgical area, medical hospitalization, surgical hospitalization, intensive care). Furthermore, we address the technique of venous catheter insertion according to relevant parameters, including the appearance of DPIVC, the number of punctures required, the patient’s perception of pain, the resources available (staff numbers and time) and the option of employing alternative methods. Finally, we examine whether access difficulty is influenced by the experience of the attending clinician, among other variables.

## 2. Materials and Methods

### 2.1. Study Design

Case-control study performed in accordance with a previously published protocol [6]. This report complies with the STROBE statement for the reporting of observational studies [18].

### 2.2. Settings

The study was conducted in medical and surgical hospitalization wards, surgical areas, A&E and intensive care units, at eight hospitals operating within the Spanish National Health System; three were university hospitals and five were second-level hospitals.

### 2.3. Participants

The study population was composed of adult patients requiring peripheral intravenous cannulation at any time and for any purpose during hospitalization. All gave written consent to participate in the study and were included prospectively. Emergency patients in live-threatening situations and pregnant women during labour were excluded.

### 2.4. Data Sources and Measurements

Clinical registered nurses working in the hospital units participating in this study selected consecutive patients for inclusion, over a period of 11 months (1 February to 31 December 2017), after receiving training on the data collection technique to be applied. During the inclusion process, the nurses were blinded to the study goals and the criteria for assignment to the case group, in order to prevent data collection bias. Within each unit, a nurse was assigned to answer questions that might arise during the data collection process. Prior to the cannulation process, the nurse recorded the variables related to his/her assessment of the situation of vascular access. On completion of the procedure, variables concerning the technique performed were also recorded. Subsequently, a researcher reviewed the clinical history of the patients included in the study, taking note of the variables related to the patient’s health and to the treatment process. This data collection system was piloted in four units within a single hospital for one month, before the study proper. The study data were collected from February to December 2017.

In this study, DPIVC is defined as follows: two or more failed punctures; the need for auxiliary techniques (ultrasound, infrared or transillumination) when accessible vessels could not be identified; the need for central access after failure to achieve peripheral access or the decision not to implement it (no venous access achieved, and abandonment of the procedure).

The case group was composed of all patients presenting with DPIVC at some time during their hospital care. The control group was composed of the remaining patients in the study population.

### 2.5. Variables

Thirteen variables were included in the analysis, derived from a previous systematic review [17]: age, vein palpability and visibility, previous history of DPIVC, upper limb alterations, previous catheters in the present episode, hospitalization or A&E attention in the last 90 days, diabetes mellitus, parenteral drug abuse, chemotherapy, extreme values of BMI (<18.5 or >30), haemodialysis and chronic obstructive pulmonary disease. The main admission diagnoses, coded by ICD-10, were also obtained. In addition, the hospital unit in which the health care was performed was also considered as a factor that might be associated with access difficulty. Regarding the healthcare professionals performing the IV cannulation, the variables considered were their total nursing experience, their experience in PIVC, their age and gender. A fuller description of these variables and of the measurement parameters applied is given in the above-mentioned protocol [6].

### 2.6. Study Size

We calculated that a minimum sample size of 2070 patients would be required, with at least 207 patients in the case group. These values were obtained using a Poisson distribution and assuming as a reference the OR of 2.1 for the risk factor ‘diabetes’ reported by Fields et al. [19], with an alpha risk of 0.05 and a beta risk of 0.2 in a bilateral contrast. Additionally, we added another ten cases for each category in each study variable using the method described by Peduzzi et al. [20].

### 2.7. Statistical Methods

For the statistical analysis, a descriptive summary was made, applying association tests between the main study variables: risk factors, characteristics of the cannulation technique, healthcare environment and variables related to the clinician’s profile.

Depending on the nature and distribution of the variables, the chi-square, Student *t*, Mann–Whitney U, Wilcoxon and Friedman W, ANOVA and Pearson tests were applied.

The potential risk factors were considered by bivariate logistic regression analysis. The variables with a statistical significance of *p <* 0.05 were included in a multivariate logistic regression model, in which the different healthcare environments were also included as predictive factors. The data were then refined by eliminating variables that did not contribute significant ORs to the model. The resulting multivariate logistic regression model thus included the independent risk factors for DPIVC. The effect size is shown with an adjusted OR and with 95% confidence intervals.

All statistical analyses were performed with SPSS 21.0.

## 3. Results

During the study period, a total of 2686 patients were recruited, and 24 of them were excluded due to incomplete recordings (*n =* 23) and revoked consent (*n =* 1). This resulted in a final sample of 2662 patients, of whom 50.3% were women. The mean age of this population was 64.3 ± 17.6 years. DPIVC criteria were presented by 221 patients, or 8.3% of the sample. The ratio of cases to controls was 1:11. Table 1 shows the general characteristics of the sample.

Bivariate analysis of the risk factors highlighted nine statistically significant variables: female sex, non-visible veins, non-palpable veins, previous history of DPIVC, acute or chronic alteration of the upper limbs (either or both), previous catheter insertion during the current hospitalization and extreme BMI. The presence of all these variables in the same person was associated with a DPIVC risk up to 30 times higher. Table 2 shows the results of this analysis.

Subsequently, statistically significant variables formed an initial predictive model of the independent risk factors for DPIVC (see Table 3), adjusted to take into account the potential influence of the care environment on the study event. In this model, the variables found to be significant were the patient’s previous history of DPIVC, the palpability of the target vein, the presence of upper limb alterations and the puncture site selected.

Table 4 shows the results obtained for the refined predictive model, from which the non-significant factors have been eliminated, together with patient records for which the puncture site was unknown. The most significant risk factor was found to be a previous history of DPIVC, which had an adjusted OR of 4.92 (95% CI 3.17 to 7.63). Performing the puncture in the forearm is a preventive factor against DPIVC, with an OR of 0.60 (95% CI 0.40 to 0.90).

The analysis according to hospital environment revealed differences in the frequency of occurrence of DPIVC and of some risk factors. Thus, statistically significant differences were found in the occurrence of upper arm alterations (more frequent in medical hospitalization units) and in the previous insertion of catheters during the current hospitalization (more frequent in medical and surgical hospitalization). Table 5 shows the factors studied by type of hospital unit.

Table 6 shows the differences in the cannulation technique used, according to the appearance or otherwise of DPIVC. Statistically significant differences were found in the number of punctures, the calibre of the catheter used, its location, the intensity of pain perceived by the patient, the number of medical professionals involved and the time elapsed until successful cannulation was achieved. The average total number of punctures made was 1.33 ± 0.76, with a range of 1 to 8.

Alternative puncture techniques or resources were only used in patients with DPIVC, which was the case for 60 patients in this group (27.1%). On 24 occasions (10.8%), the patient was referred to other medical services or professionals; in 22 cases (9.9%), ultrasound was used; on nine occasions (4.1%) a central access was inserted, although this had not previously been indicated; on five occasions (2.3%) access was obtained via a lower limb, and on thirteen occasions (5.9%) it was decided not to insert the catheter but to apply an alternative treatment.

There were no significant differences in the variables related to the medical professional performing the technique. The data for these variables are shown in Table 7.

## 4. Discussion

This study identifies independent risk factors for difficulty in peripheral venous cannulation in patients treated in a hospital setting. A previous history of DPIVC, the presence of non-palpable veins, acute alterations in the upper limbs and the selection of the antecubital fossa for puncture are all predisposing factors for DPIVC. Patients with a previous history of DPIVC are at almost five times greater risk of experiencing this difficulty again. This risk factor presented the highest OR in our sample (OR 4.92, 95% CI 3.17 to 7.63), which is in line with previous studies in this field, with various populations of hospitalized patients [19,21,22,23]. It would be very useful to systematically record this variable in the patient’s clinical history, as an alert indicator for medical professionals needing to achieve vascular access during medical treatment.

The risk factors found to be most significant were those related to the nurse’s assessment of vascular access status, which corroborates the findings of previous studies focusing on DPIVC and the efficacy of the first puncture [19,21,22,23,24]. However, we found the patient’s pre-existing health conditions to be less relevant than has been hypothesised in previous research [19,24].

The associated medical resources required, the time spent in performing this task and the number of nurses needed to do so were all greater for patients with DPIVC. Difficulty in achieving cannulation, as well as posing a health problem for the patient, produces an associated health cost, one that is frequently under-estimated [25,26]. In addition, it provokes a delay in treatment administration and is a stressful element, both for the patient and for the nurse [14,27,28].

Although venous puncture is generally considered to generate only low-intensity pain, our study confirms previous research findings that there is a clinically significant difference between the sensation of pain described by patients with DPIVC and that of patients who do not present this difficulty [29,30]. Situations of difficult access, as well as provoking multiple punctures, often result in cannulation manoeuvres that generate more tissue damage, with possible adverse effects for the patient, thus degrading the quality of healthcare and the user’s satisfaction [3,31,32].

The quality of healthcare can be significantly improved by designing interventions to improve the efficacy of the first puncture, selecting the most appropriate catheter and reducing the pain of peripheral cannulation, thus minimising its adverse effects [10,15,33]. Early referral to a vascular access specialist team (VAST) with advanced knowledge and skills in catheter insertion and care, or interventions putting into practice the recommendations of experts in the management of vascular catheters, can help to avoid the adverse events associated with cannulation, and their associated costs [13,34].

Previous studies in this event have mainly focused on specific hospital units. For this reason, our review of the literature did not reveal any prior analyses of differences in the frequency and characteristics of DPIVC according to the hospital department in which the cannulation takes place. In our sample, the area presenting the highest frequency of difficulties in vascular access was that of A&E, where 11.3% of cannulations resulted in DPIVC. Furthermore, there were significant differences in the distribution of upper limb alterations, which were most frequently observed in medical hospitalization units.

In relation to the influence of nursing skill and experience on cannulation efficacy, previous studies have suggested that there is a relationship between clinical experience and the effectiveness of the first puncture [35], although we found no evidence for this. We believe, rather, that DPIVC is a characteristic associated with the patient and with the conditions in which treatment is provided, not with the professional profile of the nurse responsible. This conclusion is in line with Rippey et al. [36], who associated clinical experience with the ability to predict difficulty, based on a prior assessment of vascular access. In addition, numerous studies have found the visibility and, above all, the palpability of the vein, according to the nurse’s assessment, to be of great significance [22,23,24,37,38]. In our sample, the non-identification of palpable veins after the application of a tourniquet was found to be an independent risk factor for DPIVC (OR 2.35, 95% CI 1.65 to 3.36).

The findings obtained in the present study are expected to be useful for nurses responsible for performing peripheral cannulation, as well as for designing, establishing and deploying VASTs. In consequence, advanced techniques might be applied in this field, reducing the adverse effects associated with catheterization [39,40]. As a final contribution, we propose a framework for the systematisation of risk factors in the clinical history of the hospitalized patient, which would enable the clinician to anticipate situations of DPIVC and forestall undesirable outcomes.

## 5. Limitations

The present study has certain limitations that should be acknowledged. According to previous research, the variable sickle cell disease is a significant risk factor for DPIVC [19]. However, this condition was not included in our analysis, due to its extreme rarity in our study population. Furthermore, the fact that the hospital staff performing the technique were responsible for compiling and providing the study data could have resulted in the mis-recording of some variables, especially the number of punctures and alternative procedures performed, as these factors may be associated with the clinician’s professional reputation. Finally, the fact that the majority of the sample derive from medical and surgical wards may influence the frequency of some variables.

## 6. Conclusions

The present study identifies four independent risk factors for DPIVC: a previous history of DPIVC, the presence of non-palpable veins and acute alterations in the upper limbs and/or the area of antecubital puncture. Significant differences in the prevalence of DPIVC were found according to the hospital area in which the intervention was performed and the resources (including time) available for cannulation. The variables related to the nurse performing the technique were not associated with the appearance of DPIVC.

## Figures and Tables

**Table 1 jcm-09-00799-t001:** General Description of the Sample.

Variable	TOTAL *n =* 2662 *n* (%) or Mean (SD)	DPIVC (Cases) *n =* 221 *n* (%) or Mean (SD)	NO DPIVC (Controls) *n =* 2441 *n* (%) or Mean (SD)	*p* Value
AGE	64.3 (17.6)	65.8 (19.1)	64.2 (17.4)	0.193
GENDER				
Female	1338 (50.3%)	136 (61.5%)	1202 (49.2%)	<0.001
Male	1324 (49.7%)	85 (38.5%)	1239 (50.8%)	
TYPE OF UNIT				
Medical hospitalization	1097 (41.2%)	85 (38.5%)	1012 (41.5%)	0.386
Surgical hospitalization	707 (26.9%)	65 (29.4%)	652 (26.7%)	0.386
A&E / Critical care	504 (18.9%)	57 (25.8%)	447 (18.3%)	0.007
Surgical area	164 (6.2%)	10 (4.5%)	154 (6.3%)	0.291
MAS	180 (6.8%)	4 (1.8%)	176 (7.2%)	0.002
WORK SHIFT				
8 a.m.–3 p.m.	1258 (47.3%)	101 (45.7%)	1157 (47.4%)	0.628
3 p.m.–10 p.m.	812 (30.5%)	83 (37.6%)	729 (29.9%)	0.017
10 p.m.–8 a.m.	592 (22.2%)	37 (16.7%)	555 (22.7%)	0.040
MAIN DIAGNOSIS / REASON FOR HOSPITAL ADMISSION				
Diseases of the digestive system (K00-K95)	522 (19.6%)	49 (22.2%)	473 (19.4%)	0.316
Diseases of the circulatory system (I00-I99)	333 (12.5%)	24 (10.9%)	309 (12.7%)	0.439
Diseases of the musculoskeletal system and connective tissue (M00-M99)	328 (12.3%)	19 (8.6%)	309 (12.7%)	0.079
Diseases of the respiratory system (J00-J99)	325 (12.2%)	26 (11.8%)	299 (12.2%)	0.833
Diseases of the genitourinary system (N00-N99)	236 (8.9%)	12 (5.4%)	224 (9.2%)	0.061
Neoplasms (C00-D49)	200 (7.5%)	12 (5.4%)	188 (7.7%)	0.220
Symptoms, signs and abnormal clinical and laboratory findings, not elsewhere classified (R00-R99)	130 (4.9%)	26 (11.8%)	104 (4.3%)	<0.001
Factors influencing health status and contact with health services (Z00-Z99)	116 (4.4%)	4 (1.8%)	112 (4.6%)	0.053
Certain infectious and parasitic diseases (A00-B99)	66 (2.5%)	5 (2.3%)	61 (2.5%)	0.829
Diseases of the skin and subcutaneous tissue (L00-L99)	60 (2.5%)	9 (4.1%)	51 (2.1%)	0.057
Diseases of the eye and adnexa (H00-H59)	57(2.1%)	3 (1.4%)	54 (2.2%)	0.401
Diseases of the blood and blood-forming organs and certain disorders involving the immune mechanism (D50-D89)	55 (2.1%)	4 (1.8%)	51 (2.1%)	0.780
Injury, poisoning and certain other consequences of external causes (S00-T88)	54 (2.0%)	7 (3.2%)	47 (1.9%)	0.210
Diseases of the nervous system (G00-G99)	48 (1.8%)	6 (2.7%)	42 (1.7%)	0.287
Endocrine, nutritional and metabolic diseases (E00-E89)	44 (1.7%)	5 (2.3%)	39 (1.6%)	0.458
Unknown / undefined	41 (1.5%)	2 (0.9%)	39 (1.6%)	0.423
Mental and behavioural disorders (F01-F99)	22 (0.8%)	3 (1.4%)	19 (0.8%)	0.363
Diseases of the ear and mastoid process (H60-H95)	18 (0.7%)	2 (0.9%)	16 (0.7%)	0.665
Pregnancy, childbirth and the puerperium (O00-O9A)	3 (0.1%)	1 (0.5%)	2 (0.1%)	0.116
Congenital malformations, deformations and chromosomal abnormalities (Q00-Q99)	2 (0.1%)	1 (0.5%)	1 (0.0%)	0.033
External causes of morbidity and mortality (V00-Y99)	2 (0.1%)	1 (0.5%)	1 (0.0%)	0.033

A&E: Accident and Emergency; MAS: Major Ambulatory Surgery.

**Table 2 jcm-09-00799-t002:** Bivariate analysis of risk factors.

Variable	Total *n =* 2663 *n* (%)	DPIVC *n =* 221 *n* (%)	NO DPIVC *n =* 2441 *n* (%)	*p*
Age >65 years	1441 (54.1)	130 (58.8)	1311 (53.7)	0.159
Female sex	1338 (50.3)	136 (61.5)	1202 (49.2)	0.001
Non-palpable veins	498 (18.)	110 (49.7)	388 (15.9)	<0.001
Non-visible veins	793 (29.8)	121 (54.7)	672 (27.5)	<0.001
History of DPIVC	1108 (41.6)	185 (83.7)	923 (37.8)	<0.001
Upper limb alterations	450 (16.9)	81 (36.6)	369 (15.1)	<0.001
Acute	320 (12.0)	51 (23.0)	269 (11.0)	<0.001
Chronic	176 (6.6)	33 (14.9)	143 (5.8)	<0.001
Previous catheters inserted during current hospitalization	1592 (59.8%)	147 (66.52)	1445 (59.2)	0.044
Admission to hospital / A&E in the last 90 days	544 (20.4)	56 (25.3)	488 (19.9)	0.068
Diabetes mellitus	586 (22.0)	50 (21.9)	536 (22.6)	0.866
Parenteral drug abuse	9 (0.3)	2 (0.9)	7 (0.2)	0.169
Chemotherapy	145 (5.4)	12 (5.4)	133 (5.4)	0.999
Haemodialysis	20 (0.8)	4 (1.8)	16 (0.7)	0.078
BMI				
<18.5	241 (9.1)	28 (13.0)	213 (9.0)	<0.001
18.5 - 30	1827 (68.6)	123 (57.2)	1704 (71.7)	
>30	523 (19.6)	64 (29.8)	459 (19.3)	
COPD	277 (10.4)	31 (14.03)	246 (10.8)	0.084

DPIVC: Difficult peripheral intravenous cannulation; BMI: Body Mass Index; COPD: Chronic obstructive pulmonary disease.

**Table 3 jcm-09-00799-t003:** Predictive model of DPIVC risk factors.

	Unadjusted		Adjusted (Puncture Site & Hospital Unit)		
	OR	95%CI	OR	95%CI	*p*
Female sex	1.65	(1.24 to 2.19)	1.10	(0.76 to 1.6)	0.600
Non- palpable veins	5.24	(3.94 to 6.97)	2.74	(1.85 to 4.06)	0.000*
Non- visible veins	3.19	(2.41 to 4.21)	1.28	(0.86 to 1.89)	0.221
History of DPIVC	8.45	(5.86 to 12.19)	3.53	(2.17 to 5.73)	0.000*
Upper limb alterations	3.25	(2.42 to 4.36)	1.24	(0.51 to 3.01)	0.634
Acute upper limb alterations	2.42	(1.73 to 3.4)	1.62	(0.67 to 3.91)	0.281
Chronic upper limb alterations	2.82	(1.88 to 4.24)	1.31	(0.57 to 3.02)	0.526
Previous catheters inserted during current hospitalization	1.37	(1.02 to 1.83)	0.86	(0.51 to 1.46)	0.574
BMI >30	1.88	(1.42 to 2.49)	1.04	(0.7 to 1.54)	0.862
Puncture site: Antecubital fossa	-	-	1.84	(1.14 to 2.95)	0.012*
Puncture site: Forehand	-	-	0.66	(0.41 to 1.04)	0.075
Unit: Medical ward	-	-	0.53	(0.17 to 1.58)	0.254
Unit: Surgical ward	-	-	1.10	(0.37 to 3.23)	0.861
Unit: A&E / Critical care	-	-	1.09	(0.38 to 3.13)	0.870
Unit: Surgical area / MAS	-	-	0.54	(0.13 to 2.17)	0.385

DPIVC: Difficult peripheral intravenous cannulation; BMI: Body Mass Index; A&E: Accident and Emergency; MAS: Major Ambulatory Surgery.

**Table 4 jcm-09-00799-t004:** Refined predictive model of DPIVC risk factors.

	Crude Data		Refined			
	OR	95% CI	B	OR	95% CI	*p*
History of DPIVC	8.45	(5.86 to 12.19)	1.59	4.92	(3.17 to 7.63)	<0.001
Non-palpable veins	5.24	(3.94 to 6.97)	0.86	2.35	(1.65 to 3.36)	<0.001
Acute upper limb alterations	2.42	(1.73 to 3.4)	0.44	1.56	(1.06 to 2.30)	0.024
Puncture site: antecubital fossa	3.48	(2.39 to 5.07)	0.61	1.84	(1.23 to 2.75)	0.030
Puncture site: forehand	0.29	(0.2 to 0.42)	−0.51	0.60	(0.40 to 0.90)	0.014

Hosmer-Lemeshow: χ^2^: 6.58; *p =* 0.254; R^2^Nagelkerke: 0.20. DPIVC: Difficult peripheral intravenous cannulation.

**Table 5 jcm-09-00799-t005:** Characteristics of the patients with DPIVC according to hospitalisation unit.

	Medical *n =* 1097 *n* (%)	Surgical *n =* 717 *n* (%)	A&E *n =* 504 *n* (%)	Operating Theatre *n =* 180 *n* (%)	MAS *n =* 164 *n* (%)	*p*
Patients with DPIVC	85 (7.7%)	65 (9.1%)	57 (11.3%)	10 (6.1%)	4 (2.2%)	0.002
Age >65 years	54 (63.8%)	38 (58.5%)	32 (56.1%)	3 (30.0%)	3 (75.0%)	0.307
Female sex	56 (65.9%)	38 (58.5%)	37 (64.9%)	2 (20.0%)	3 (75.0%)	0.067
Non-palpable veins	44 (51.8%)	31 (47.7%)	31 (55.4%)	3 (33.3%)	1 (25.0%)	0.578
Non-visible veins	44 (51.8%)	39 (60%)	32 (58%)	6 (66.7%)	0 (0%)	0.158
History of DPIVC	75 (88.2%)	50 (76.9%)	50 (87.7%)	8 (80.0%)	2 (50.0%)	0.108
Upper limb alterations	48 (62.3%)	20 (30.8%)	10 (21.7%)	3 (37.5%)	0 (0%)	<0.001
Acute	39 (52.0%)	12 (18.5%)	9 (19.1%)	1 (16.7%)	0 (0%)	<0.001
Chronic	16 (22.5%)	8 (12.5%)	8 (17.4%)	1 (16.7%)	0 (0%)	0.520
Previous catheters inserted during current hospitalization	73 (85.9%)	59 (90.8%)	13 (22.8%)	1 (10.0%)	1 (25.0%)	<0.001
Admission to hospital / A&E in the last 90 days	21 (24.7%)	12 (18.5%)	21 (37.5%)	1 (10.0%)	1 (25.0%)	0.124
Diabetes mellitus	18 (21.2%)	15 (23.1%)	14 (24.6%)	3 (30.0%)	0 (0%)	0.789
Parenteral drug abuse	0 (0%)	1 (1.5%)	1 (1.8%)	0 (0%)	0 (0%)	0.799
Chemotherapy	4 (4.7%)	4 (6.2%)	4 (7.0%)	0 (0%)	0 (0%)	0.872
Haemodialysis	0 (0%)	2 (3.1%)	2 (3.5%)	0 (0%)	0 (0%)	0.508
BMI						0.438
<18.5	15 (18.3%)	6 (9.2%)	6 (10.9%)	1 (11.1%)	0 (0%)	
18.5 - 30	43 (52.4%)	42 (64.6%)	29 (52.7%)	5 (55.6%)	4 (100%)	
>30	24 (29.3%)	17 (26.2%)	20 (36.4%)	3 (33.3%)	0 (0%)	
COPD	12 (14.1%)	13 (20.0%)	4 (7.0%)	2 (20.0%)	0 (0%)	0.268

DPIVC: Difficult peripheral intravenous cannulation; BMI: Body Mass Index; COPD: Chronic obstructive pulmonary disease; A&E: Accident and Emergency; MAS: Major Ambulatory Surgery.

**Table 6 jcm-09-00799-t006:** Cannulation technique according to difficulty.

Variable	Total *n* (%) or Mean (SD)	DPIVC *n* (%) or Mean (SD)	NO DPIVC *n* (%) or Mean (SD)	*p*
SBP before cannulation	126.8 (20.4)	125.4 (21.7)	128.0 (21.2)	0.075
DBP before cannulation	70.6 (12.3)	69.6 (12.9)	71.4 (12.9)	0.151
Number of punctures	1.3 (0.7)	3.3 (1.1)	1.15 (0.3)	<0.001
First attempt success	2087 (78.39%)	2074 (85.00%)	13 (5.90%)	
Calibre of catheter inserted				
14 gauge	1 (0.1%)	0 (0%)	1 (0.1%)	<0.001
16 gauge	6 (0.2%)	0 (0%)	6 (0.2%)	
18 gauge	659 (24.9%)	26 (12.3%)	633 (25.9%)	
20 gauge	1506 (56.8%)	96 (45.5%)	1410 (57.8%)	
22 gauge	428 (16.1%)	71 (33.6%)	357 (14.6%)	
24 gauge	43 (1.6%)	10 (4.7%)	33 (1.4%)	
CANNULATION SITE				
Forehand	1216 (45.7%)	52 (23.5%)	1164 (47.7%)	<0.001
Hand	841 (31.6%)	69 (31.2%)	772 (31.6%)	
Antecubital fossa	513 (19.3%)	69 (31.2%)	444 (18.2%)	
Not recorded	92 (3.5%)	31 (14.0%)	61 (2.5%)	
Registered nurses participating (n)	1.3 (0.7)	1.6 (0.7)	1.1 (0.3)	<0.001
Nursing assistants participating (n)	0.1 (0.3)	0.4 (0.5)	0.2 (0.4)	<0.001
Time to cannulation (minutes)	8.8 (7.1)	24.0 (13.5)	7.6 (5.3)	<0.001
Pain intensity (Visual Analogue Scale)	1.8 (1.9)	3.0 (2.6)	1.8 (1.9)	<0.001

DPIVC: Difficult peripheral intravenous cannulation; SBP: Systolic blood pressure; DBP: Diastolic blood pressure.

**Table 7 jcm-09-00799-t007:** Variables related to the nurses performing the cannulation.

Variable	Total *n* (%) or Mean (SD)	DPIVC *n* (%) or Mean (SD)	NO DPIVC *n* (%) or Mean (SD)	*p*
Age	39.6 (6.9)	39.1 (7.1)	39.1 (6.2)	0.893
Sex				
Female	2086 (83.1%)	155 (78.7%)	1931 (83.5%)	0.082
Male	423 (16.9%)	42 (21.3%)	381 (16.5%)	
Nursing experience (years)	16.3 (6.4)	15.6 (7.0)	15.6 (6.7)	0.758
PIVC experience (years)	14.5 (6.5)	13.6 (7.1)	13.8 (6.6)	0.838

DPIVC: Difficult peripheral intravenous cannulation.

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
