# Peer review of "Risk Factors for Difficult Peripheral Intravenous Cannulation. The PIVV2 Multicentre Case-Control Study"

_jcm, 2020, doi:10.3390/jcm9030799_

Round 1
Reviewer 1 Report
This research project was performed on an interesting and common issue in medical healtcare: peripheral intravenous vascular access. Congratulations with the study and its results. I suggest some revisions or clarifications.
Introduction
The introduction section provides sufficient background information about the subject to be studied and highlights the aim of the study. Despite, I suggest some textual and grammatical adjestments:
- Alinea 2, line 3: change "reducing" into "reduce".
- Alinea 2, line 5: change "difficulty in peripheral cannulation" into "difficulty to obtain peripheral vascular access".
- Alinea 4, line 6: it's difficult to understand the sentence "the first peripheral puncture and select a suitable catheter at the start of intravenous treatment", I suggest to revise this to make clear what the author means with it.
- Alinea 5, line 1: remove "the".
- Alinea 5, line 1: please specify who were meant with "us".
- Alinea 7, line 2: the abbreviation (A&E) must be completely written out on the first time it is used, with the abbreviation placed between brackets. For me, it is not clear what A&E stands for.
Methods
The description of the participants included in this study is not completely clear to me. For what reason were emergency patients and pregnant women in the final stage of childbirth excluded from this trial? Please give a discription for this.
The authors describe that consecutive patients were selected by nurses who participated in this study and collected date. Later on, authors mention that the nurses were blinded to the study goal in order to prevent selection bias. In my opinion, it sounds strange that selection of patients will result in the prevention of selection bias. For what reason did you not use (for instance) a non-probability consecutive sampling technique? I suggest to rephrase the section about selection of participants to make the applied strategy clear.
Authors included thriteen variables into the analyses. On the basis of which were these variables chosen?
It is not clear to me how the minumum sample size of 2070 patients was calculated. It is clear to me that 10 patients need to be included for each case in this trial. Can you please explain the used technique to calculate the sample size more completely? And you please explain what the added value is of using the risk factor diabetes from a study by Fields et al.?
Please add to the statistical analyses section which program was used for all statistical analyses.
Results
In general, I think the presentation of the results section can be displayed much clearly, by simply follow the plan as descriped in the statistical analyses section. In my opninion, the presentation of the results do not match the presented analyses technique.
A few simple questions need to be answered at the beginning of the results section:
- What was the first attempt success rate for peripheral intravenous cannulation throughout the total cohort, and in the subgroups?
- How many patients were included in total, and how many of them were excluded from final analyses and for what reason?
- For the analyses in Table 2, a bivariate technique was used. This was not mentioned in the statistical analyses section. Why did the authors use this technique instead of the described univariate logistic regression analyses?
- Regarding Table 3, the authors described in the text that the palpability of the target vein was found to be significant, although a P value of 0.221 was represented in Table 3. Can you please explain on which occasion the palpability of the vein was found to be significant?
- To continue, the significant variables from Table 3 were included in the analyses for Table 4 (including palpability of the target vein). Can you please explain this apparent inaccuracy?
- For a clear presentation of the results according to the represented Tables, I suggest to use the same order of variables each time (Table 2 and Table 5 have a different order of variables, for instance).
Discussion
It is interesting that you mention the use and/or implementation of a vascular access specialt team (VAST), although a Cochrane Review could not support of refute that the assertion of a VAST is superior to the generalist model for device insertion and prevention of failure (Carr et al. 2018). Can the authors please explain why a VAST should be created and what is its impact on vascular access device insertion and maintance?
The author mention that previous studies suggested that there is a relationship between clinical experience and the effectiveness of the first puncture. Although this seems trivial, the study by Van Loon et al. (2019) did not represent this results, despite the authors used this reference here (reference number 36). Please explain or correct this inaccuracy.
Limitations
A few simple questions need to be answered at the beginning of the limitations section:
- Is it correct that the author mentioned "spindle cell disease" (which is a type of connective tissue cancer), or do you mean "sickle cell disease"?
- The vast majority of patients were recruited from the medical department, followed by the surgical department (73%). Can you describe whether or not this influeces the study results?
- Was there some other bias (information or recall) or confounding in this study, and how did the authors handle this?
Reviewer 2 Report
Dear authors,
Congratulations on your manuscript!
Overall, it's well written, structured and fine for publication. Nevertheless, I deem it necessary to note some things for your consideration and reflection (mostly just questions for discussion):
INTRODUCTION
1) I'm grateful to read in the Introduction a cost-effectiveness data ("cost savings of 220-1600 US"). Congrats.
2) I suggest that you define A&E when it first appears in the text (Introduction, 8th paragraph).
MATERIALS AND METHODS
1) In the protocol published in BMJ in 2019, you specify the study period (1 February to 31 December 2017). I suggest the inclusion of this data right after you mention the 11 months of patient selection period. Or the periods are different? It becomes confusing, since the patients are included prospectively.
2) I missed an explanation of how do you treat patients that already had a catheter when they were admitted to the hospital wards or that were submitted to catheterization when they had to go to surgery or a complementary exam. They opt-out? Or, even if they are from that ward, you simply don't consider them, since you want to testify catheter insertion?
3) The variables considered for healthcare professionals were nursing experience, PIVC experience, age and gender. Wouldn't specific PIVC formation (courses, advanced workshops) be a relevant variable?
DISCUSSION
1) I suggest the definition of VASTs in the 8th paragraph.
